# AI-Enhanced Lower Extremity X-Ray Segmentation: A Promising Tool for Sarcopenia Diagnosis

**DOI:** 10.3390/healthcare13192488

**Published:** 2025-09-30

**Authors:** Hyunwoo Park, Hyeonsu Kim, Junil Yoo

**Affiliations:** 1Department of Internal Medicine, Hallym University Medical Center, Hallym University College of Medicine, Anyang 14068, Republic of Korea; andyr2d2@hallym.or.kr; 2Department of Biomedical Research Institute, Inha University Hospital, Incheon 22188, Republic of Korea; lemonjames96@gmail.com; 3Department of Orthopedic Surgery, Inha University Hospital, Inha University College of Medicine, Incheon 22188, Republic of Korea

**Keywords:** deep learning, semantic segmentation, X-ray, thigh, skeletal muscle

## Abstract

**Background/Objectives**: Sarcopenia, characterized by progressive loss of skeletal muscle mass and strength, significantly impacts physical function and quality of life in older adults. Traditional measurement methods like Dual-energy X-ray absorptiometry (DEXA) are often inaccessible in primary care. This study aimed to develop and validate an AI-driven auto-segmentation model for muscle mass assessment using long X-rays as a more accessible alternative to DEXA. **Methods**: This was a retrospective validation study using data from the Real Hip Cohort at Inha University Hospital in South Korea. 351 lower extremity X-ray images from 157 patients were collected and analyzed. AI-based semantic segmentation models, including U-Net, V-Net, and U-Net++, were trained and validated on this dataset to automatically segment muscle regions. Model performance was assessed using Intersection over Union (IoU) and Dice Similarity Coefficient (DC) metrics. The correlation between AI-derived muscle measurements and the DEXA-derived skeletal muscle index was evaluated using Pearson correlation analysis and Bland–Altman analysis. **Results**: The study analyzed data from 157 patients (mean age 77.1 years). The U-Net++ architecture achieved the best segmentation performance with an IoU of 0.93 and DC of 0.95. Pearson correlation demonstrated a moderate to strong positive correlation between the AI model’s muscle estimates and DEXA results (r = 0.72, *** *p* < 0.0001). Regression analysis showed a coefficient of 0.74, indicating good agreement with reference measurements. **Conclusions**: This study successfully developed and validated an AI-driven auto-segmentation model for estimating muscle mass from long X-rays. The model provides an accessible alternative to DEXA, with potential to improve sarcopenia diagnosis and management in community and primary care settings. Future work will refine the model and explore its application to additional muscle groups.

## 1. Introduction

Sarcopenia, characterized by progressive loss of skeletal muscle mass and strength, has emerged as one of the most critical chronic conditions in recent decades, with its significance extending far beyond mere physical decline to profoundly impact quality of life, morbidity of chronic diseases, and remaining life expectancy of elderly populations [1,2,3]. As global aging populations continue to expand, sarcopenia has evolved from a recognized aging phenomenon into a major public health challenge, affecting 10–27% of individuals over 60 years and up to 30% of those over 65 years, though prevalence varies considerably across countries and diagnostic criteria [4]. For instance, South Korea reports prevalence rates of 6.6% in men and 10.3% in women aged 65 and older, while rural areas consistently show higher rates than urban regions due to differences in lifestyle, nutrition, and healthcare accessibility [5]. The socioeconomic burden is staggering, with direct healthcare costs reaching approximately $18.5 billion annually in the United States and £2.5 billion in the United Kingdom, translating to additional costs of $2315.7 per patient in the US and £2707 per patient in the UK [6]. This economic impact is particularly pronounced among socioeconomically disadvantaged populations, who experience higher incidence rates and reduced access to early prevention and treatment, thereby exacerbating health inequalities [7].

Over the past 30 years, continental consensus groups have developed and refined diagnostic criteria, culminating in the recent Global Leadership Initiative on Sarcopenia (GLIS) guidelines, which have narrowed the focus to precise muscle mass measurement and specific muscle strength assessments [8]. However, despite these advances, significant challenges persist in clinical practice, including lack of standardization across different diagnostic criteria, insufficient availability of specialized equipment and trained personnel for accurate muscle mass and strength evaluation, and limited awareness among healthcare providers regarding the importance of early sarcopenia detection and management [9]. While effective management requires multidisciplinary approaches integrating nutrition, exercise, and pharmacological interventions, the fundamental barrier lies in the inconsistent and often inadequate diagnostic practices that prevent timely identification and appropriate therapeutic intervention [10,11].

Previous guidelines, such as those from the Asian Working Group for Sarcopenia (AWGS 2019) and the European Working Group for Sarcopenia in Older People (EWGSOP2), established standards for identifying and assessing sarcopenia based on three aspects: muscle mass (skeletal muscle index), muscle strength (hand grip strength), and physical performance such as gait speed and short physical performance battery (SPPB) [12,13]. However, in the process of developing the GLIS guidelines, sarcopenia has been defined in terms of muscle mass, muscle strength, and muscle-specific strength (muscle strength/muscle size), underscoring the critical importance of accurate skeletal muscle mass measurement. Currently, the gold standard methods for measuring muscle mass include whole-body Dual-energy X-ray absorptiometry (DEXA), CT and MRI [14,15,16]. However, these advanced imaging techniques present significant challenges in primary care settings due to their limited accessibility [17]. While Bioelectrical Impedance Analysis (BIA) could be an alternative, its lack of visual representation of muscle tissue limits its ability to evaluate the effectiveness of specific exercise interventions or treatments based solely on numerical changes [18,19]. Therefore, the need for more accessible, user-friendly diagnostic devices, such as X-ray and ultrasound, is being emphasized, especially considering that chronic conditions like sarcopenia are ideally managed at the community and primary care settings.

Recent advances in artificial intelligence have revolutionized medical imaging analysis for muscle mass assessment, particularly in CT and MRI interpretation. Studies have demonstrated that AI-based approaches offer unique advantages including elimination of inter-observer variability that commonly affects manual measurements, capability to detect subtle radiographic patterns invisible to human observers, standardization of measurement protocols across different healthcare settings, and potential for continuous learning through exposure to diverse patient populations [20]. Recent research published in Scientific Reports has demonstrated the effectiveness of U-net transformer architecture for precise individual muscle segmentation in whole thigh CT scans for sarcopenia assessment, achieving superior accuracy in muscle boundary delineation and quantification compared to conventional manual methods [21].

Given these promising developments in AI-assisted muscle assessment and the pressing need for more accessible diagnostic tools in primary care, X-ray imaging emerges as an ideal candidate for AI-driven muscle analysis due to its widespread accessibility in primary healthcare settings. Therefore, the purpose of this study is to develop and validate a novel AI-driven automated segmentation model specifically designed for muscle mass assessment using whole-body X-rays, with the aim of establishing a standardized imaging protocol and demonstrating clinical validity through direct comparison with gold standard DEXA measurements.

## 2. Methods

### 2.1. Introduction

This study aimed to develop and validate a robust AI-based semantic segmentation model for accurately quantifying muscle mass in the thigh and calf regions from lower extremity X-ray images. Leveraging a comprehensive dataset comprising 351 X-ray images and 66 paired Dual-Energy X-ray Absorptiometry (DEXA) scans from the Real Hip Cohort in South Korea, the methodology involved the development and training of advanced deep learning architectures, specifically U-Net, V-Net, and U-Net++, on expertly annotated and pre-processed images. The developed model’s performance was rigorously evaluated using various metrics, including Intersection over Union (IoU), Dice Coefficient (DC), Average Distance (AD), and Hausdorff Distance (HD), with its accuracy and reliability subsequently validated through statistical comparisons, such as Pearson’s correlation coefficient and BlanRd-Altman analysis, against DEXA-derived skeletal muscle index measurements. All research procedures strictly adhered to the principles of the Declaration of Helsinki and received approval from the Institutional Review Board of Inha University Hospital (IRB No. INUH 2023-04-027).

### 2.2. Study Design

This study developed a robust AI-based semantic segmentation model to accurately measure muscle groups in the thigh and calf regions. Furthermore, a comparative study against Dual-Energy X-ray Absorptiometry (DEXA) scans was conducted to validate the accuracy and reliability of the AI-based semantic segmentation model for quantifying muscle mass. The training process for this model utilized a dataset comprising 351 lower extremity X-ray images from the Real Hip Cohort, a comprehensive hip fracture cohort based in South Korea.

This study adhered to the principles of the Declaration of Helsinki and was approved by the Institutional Review Board (IRB No. INUH 2023-04-027) at Inha University Hospital. All research procedures were carried out with strict adherence to ethical standards, including protection of participant privacy, confidentiality, and rights.

### 2.3. Study Setting and Participants

The study assembled a cohort of 157 patients from the Real Hip Cohort at Inha University Hospital in South Korea. This cohort provides a comprehensive repository of medical imaging data and clinical information. Specifically, 351 lower extremity X-ray images and 66 Dual-Energy X-ray Absorptiometry (DEXA) scans were collected from these individuals.

Initially, 878 individuals were identified, from whom 157 had lower extremity X-ray data (351 images). Among these, 351 images were ultimately used for semantic segmentation (281 for training and 70 for validation), after excluding 72 single-leg rotation images and 60 external rotation images due to rotation. For outcome analysis, 62 individuals provided 66 paired DEXA scans and X-ray images, and 1-year follow-up data from 4 individuals (8 paired DEXA scans and X-ray images) were also included for further discussion. The specific criteria for this study and the patient selection flowchart are detailed in Figure 1.

### 2.4. Data Acquisition and Ground Truth Labeling

Each participant’s lower extremity X-ray images were taken in a standing position using a X-ray machine (Siemens Healthineers, Erlangen, Germany). These images capture the entire lower extremities, spanning from the pelvic region down to the ankles, and are stored in the industry-standard DICOM file format. Participants whose lower extremities were externally rotated were excluded from the training process, as this rotation could affect the accuracy of the AI-based segmentation model by causing changes in distinct regions (lateral, medial, and calf) of the X-ray, particularly affecting the lower leg.

Additionally, Dual-Energy X-ray Absorptiometry (DEXA) scans were performed under a standard protocol using a GE Lunar DXA machine (GE Healthcare, Madison, WI, USA). These scans provided a detailed assessment of bone density and body composition, including detailed measurements of fat mass (g) and lean mass (g) for various body regions like arms, legs, and trunks.

To establish ground truth labels for training the AI model, we employed an annotation tool provided by 3D Slicer software (version 5.6.2). This allowed for precise delineation of various anatomical structures and tissue types. As displayed in Figure 2, we segmented the muscles into three distinct classes: lateral, medial, and gluteal regions within the thigh, as well as the calf region. Furthermore, we annotated subcutaneous fat and bony structures such as the femur, tibia, and iliac bones. These meticulously labeled annotation masks were saved in the NRRD file format, which incorporates crucial information regarding pixel spacing in real-world units, enabling accurate quantification and analysis.

For the manual segmentation, a standardized protocol was meticulously followed, utilizing anatomical landmarks and boundary definitions established based on the Netter Atlas of Human Anatomy. This rigorous approach ensured consistency and accuracy across all images. The segmentation process incorporated a rigorous three-tier quality control procedure: initially, the first author (H. Park, Department of Internal Medicine) performed the primary manual segmentation of all images. Subsequently, a co-author (H. Kim, Department of Biomedical Research Institute) independently double-checked all segmented images for accuracy and strict adherence to the anatomical guidelines. This co-author brings extensive experience in developing AI models for muscle mass measurement using lower extremity CT scans [21]. Finally, the corresponding author (J. Yoo, Department of Orthopedic Surgery), a board-certified orthopedic specialist with profound expertise in musculoskeletal anatomy and imaging, provided final confirmation for all segmented images. This structured approach, leveraging multiple expert reviews, was instrumental in ensuring the high quality, consistency, and clinical relevance of the ground truth segmentation datasets for AI model training.

### 2.5. Deep-Learning Methods of Semantic Segmentation Model

#### 2.5.1. Pre-Processing and Augmentation

In order to augment the relatively limited training dataset, we randomly flipped the image horizontally. This augmentation technique helps increase the diversity of the training data and improve the model’s ability to generalize. The images were resized to (512, 1024) Numpy Shape for training the model, ensuring consistent input dimensions for the U-Net based neural network.

Additionally, we applied several pre-processing steps using image transforms to segments muscle tissue, subcutaneous fat and skeletal bone:Scale Intensity (min = 0.0, max = 1.0): This transform scales the intensity values of the image to a range between 0.0 and 1.0. Normalizing the intensity range helps standardize the input data across different images, which can improve the model’s learning process and performance.Histogram Normalize: Histogram normalization is applied to enhance image contrast. This technique redistributes the intensity values of the image to convert the full range of possible intensities more evenly. It can help in making features more distinguishable, especially in medical imaging where contrast can be critical for identifying structures.Normalize Intensity: Standardized input data by subtracting the mean and dividing by the standard deviation of image intensities, aiding faster convergence and improved performance.Adjust Contrast (gamma = 1.6): Gamma correction was used to increase contrast, making darker regions darker and brighter regions brighter, which enhances image features.

As shown in Figure 3, these pre-processing steps work together to standardize the input data, enhance important features and improve the overall quality of the images before they are fed into the neural network for training. Crucially, all these pre-processing procedures, including image resizing and random horizontal flipping, were rigorously standardized. Each step was applied consistently to every image using the same predefined algorithms and parameters (e.g., Scale Intensity to min = 0.0, max = 1.0; Adjust Contrast with gamma = 1.6). This entire process was fully automated and executed via programmatic scripts, ensuring an entirely objective approach without any human intervention or subjective judgment.

#### 2.5.2. Model Architectures

For the AI-based automatic segmentation model, we employed U-Net, V-Net, and U-Net++ architectures. U-Net is known for its U-shaped encoder–decoder structure with skip connections, making it effective for biomedical image segmentation by combining low-level and high-level features for precise object boundary localization [22]. V-Net extends U-Net for 3D image segmentation, utilizing 3D convolutional layers and incorporating residual learning to facilitate the training of deeper networks, with strided convolutions for downsampling to preserve spatial information [23]. U-Net++, an advanced version of U-Net, further improves segmentation accuracy through nested and dense skip connections that reduce the semantic gap between encoder and decoder feature maps, along with deep supervision to enhance gradient flow [24]. These architectures were chosen for their proven capabilities in various medical imaging segmentation tasks.

#### 2.5.3. Model Training

Training was conducted on high-performance workstations equipped with NVIDIA RTX 4090 and RTX A6000 GPUs, operating on an Ubuntu 20.04 platform. The deep learning-based segmentation model was implemented using PyTorch version 2.4.1 (https://pytorch.org/) and MONAI version 1.3.2 (https://monai.io/) frameworks.

The Dice coefficient was adopted as the loss function, a frequent choice in semantic segmentation. The model was fine-tuned using the AdamW optimizer with the following hyperparameters: learning rate: 8 × 10^−5^, weight decay: 1 × 10^−5^, batch size: 8, and 3000 epochs. These hyperparameters were determined through a grid search methodology, exploring learning rates from 1 × 10^−3^ to 1 × 10^−5^, with the final configuration selected via heuristic techniques due to its superior performance on the muscle segmentation datasets.

### 2.6. Performance Evaluation and Statistical Analysis

To evaluate our model’s performance comprehensively, we employed a range of metrics: (i) Intersection over Union (IoU), (ii) Dice Coefficient (DC), (iii) Average Distance (AD), (iv) Hausdorff Distance (HD), and (v) Relative Absolute Area Difference (RAAD).

The IoU and DC provide clear indicators of overlap accuracy between the predicted and ground truth segments, essential for assessing overall segmentation quality. The AD measures the mean spatial discrepancy between these boundaries, highlighting the model’s precision in contour delineation. The HD focuses on the maximum boundary discrepancy, emphasizing the model’s capability to capture the most significant boundary deviations, which is critical for precise applications. Finally, the RAAD evaluates the proportional difference in area, offering insight into the model’s consistency in estimating the size of segmented regions. Each metric contributes unique insights and addresses specific aspects of the segmentation process, facilitating a thorough and nuanced assessment of model performance.

(i)Intersection over Union (IoU) for Multiclass

For each class *c* within the total number of classes n, the IoU was calculated by identifying the number of pixels where both the prediction and the ground truth label were equal to c (intersection), and the number of pixels where either the prediction or the ground truth label was c (union). The IoU score for each class c was computed using the formula(1)IoUc=Σpred∧labelΣPred∨label
where pred and label are binary masks indicating the presence of class *c* in the predicted and ground truth label images, respectively. If the union of the predicted and ground truth pixels for a class was zero, the IoU was set to 1, indicating a perfect match. The overall performance was quantified by averaging the IoU scores across all classes:(2)$$\mathrm{Mean}\;\mathrm{IoU}=\frac{1}{n}\sum_{c=0}^{n-1}{\mathrm{IoU}}_{c}$$

(ii)Dice Coefficient (DC) for Multiclass

The Dice Coefficient, another metric to assess segmentation accuracy, was calculated for each class c by determining the number of pixels where both the predicted and ground truth labels were c (intersection) and the total number of pixels predicted as c or labeled as c (union). The Dice score for each class was calculated as follows:(3)$$\mathrm{Dic}e_{c}=\frac{2\times \Sigma \left(\mathrm{pred}\cdot \mathrm{label}\right)}{\Sigma \left(\mathrm{pred}\right)+\Sigma \left(\mathrm{label}\right)}$$

The final Dice Coefficient across all classes was computed by averaging the individual class scores:(4)$$\mathrm{Mean}\;\mathrm{Dice}=\frac{1}{n}\sum_{c=0}^{n-1}{\mathrm{Dice}}_{c}$$

(iii)Average Distance (AD) for Multiclass

The Average Distance (AD) metric was utilized to quantify the spatial discrepancy between predicted and ground truth segmentations. For each class c, the AD was determined by calculating the mean distance of each pixel in the predicted mask to the nearest pixel in the ground truth mask and vice versa. This was performed using the Euclidean distance transform on the complements of the binary masks:(5)$$\mathrm{AvgDist}\_c=\mathrm{mean}(\mathrm{pred}\left[\mathrm{label}\right])+\frac{\mathrm{mean}\left(\mathrm{label}\left[\mathrm{pred}\right]\right)}{2}$$

If both pred class and label class were empty, the average distance was set to 0. In cases where one was empty and the other was not, the distance was considered infinite. The overall AD was averaged across all classes:(6)$$\mathrm{Mean}\;\mathrm{AvgDist}=\frac{1}{n}\sum_{c=0}^{n-1}{\mathrm{AvgDist}}_{c}$$

(iv)Relative Absolute Area Difference (RAAD) for Multiclass

The Relative Absolute Area Difference (RAAD) metric measured the percentage difference in area between the predicted and ground truth segments for each class c. This was calculated by determining the absolute difference between the areas of predicted and labeled pixels, normalized by the labeled area:(7)$${\mathrm{RelAbsDiff}}_{c}=\left|\frac{{\mathrm{pred}}_{\mathrm{area}}-{\mathrm{label}}_{\mathrm{area}}}{{\mathrm{label}}_{1\mathrm{abd}}}\right|\times 100\%$$
where pred_area_ and label_area_ denote the number of pixels predicted and labeled as class c, respectively. In instances where the label_area_ was zero, the RAAD was set to 0 if pred_area_ was also zero (indicating no discrepancy), or 100% otherwise. The final score was computed as a weighted average based on class areas:(8)$$\mathrm{Weighted}\;\mathrm{Avg}\;\mathrm{RelAbsDiff}=\frac{\sum_{c=0}^{n-1}{\mathrm{RelAbsDiff}}_{c}\times {\mathrm{class}}_{{\mathrm{area}}_{c}}}{{\mathrm{total}}_{\mathrm{area}}}$$

(v)Hausdorff Distance (HD) for Multiclass

The Hausdorff Distance (HD) metric was employed to measure the maximum boundary distance between predicted and ground truth segmentations for each class c. The HD for each class was computed by finding the maximum distance from any point in the predicted segmentation to the nearest point in the ground truth segmentation and vice versa:(9)$$Haus〖Dorff〗\_c=max(directed\_hausdorff)\left(pred_{points},label_{points}\right),directed_{hausdorff\left(label_{points},pred_{points}\right)}$$
where pred_points_ and label_points_ represent the coordinates of pixels in the predicted and ground truth masks for class c, respectively. If both pred and label were empty, the HD was considered 0. If one was empty and the other was not, the HD was set to infinity. The final Hausdorff distance was averaged across all classes:(10)$$Mean\;Hausdorff=\frac{1}{n}\sum_{c=0}^{n-1}{Hausdorff}_{c}$$

### 2.7. Comparison with Dual-Energy X-Ray Absorptiometry

#### 2.7.1. DEXA Measurements

Dual-Energy X-ray Absorptiometry (DEXA) scans were performed on a subset of 66 subjects using Lunar Prodigy Advance DEXA machine. The skeletal muscle index was calculated from these scans according to standard protocols, defined as the lean mass (arms and legs) divided by the square of the subject’s height (kg/m^2^). DEXA measurements were conducted within [time frame] of the X-ray imaging to ensure comparability.

#### 2.7.2. Comparison of Segmentation Results with DEXA

The muscle area derived from the segmentation models was compared with the DEXA skeletal muscle index. Muscle area from segmentation was calculated by summing the pixels classified as muscle and converting them to physical area using known pixel dimensions. Statistical analysis was performed using Python (version 3.9) with the SciPy and statsmodels libraries.

Pearson correlation analysis was conducted to assess the linear relationship between the segmentation-derived muscle area and the DEXA skeletal muscle index. Bland–Altman plots were generated to visualize the agreement between the two measurement methods, showing the mean difference and 95% limits of agreement. Paired t-tests were performed to determine if there were significant differences between the segmentation-based estimates and DEXA measurements.

## 3. Results

### 3.1. Demographic Characteristics

The study included 157 patients, with a demographic breakdown of 117 females and 40 males, and an average age of 77.1 years (77.16 years for females and 77.04 years for males). To evaluate the correlation between segmented muscle regions and skeletal muscle in the lower limbs, we utilized a total of 66 lower extremity X-ray images and 66 DEXA scans from 62 patients. Among the 66 DEXA results, 19 patients were classified as non-sarcopenic, 8 as having sarcopenia, and 39 as having severe sarcopenia.

A detailed overview of the participants’ demographic data, including the number of participants, age, height, and weight (mean ± SD) for female, male, and total participants across non-sarcopenia, sarcopenia, and severe sarcopenia groups (as determined by DEXA measurements), is presented in Table 1. In addition to imaging data, the Real Hip Cohort provides extensive medical data, including physical performance metrics such as the Short Physical Performance Battery (SPPB), Koval’s score, grip strength, and gait analysis data, which allowed for a comprehensive evaluation of the participants’ musculoskeletal condition.

### 3.2. Semantic Segmentation Model Performance Evaluation

For the semantic segmentation model training, we employed 351 lower extremity X-ray images. As displayed in Table 2, our proposed model, which is based on the U-Net++ architecture, demonstrated superior performance in comparison to other models such as U-Net and V-Net. Specifically, the U-Net++ model achieved an Intersection over Union (IoU) of 0.93 and a Dice coefficient (DC) of 0.95, indicating a high degree of overlap between the predicted and ground truth segmentations.

Furthermore, U-Net++ also recorded the lowest Average Distance (AD) at 0.89 ± 0.021, a Hausdorff Distance (HD) of 1.92 ± 0.101, and a Relative Absolute Area Difference (RAAD) of 1.80 ± 0.019% among the evaluated models. These metrics collectively highlight U-Net++’s precise boundary delineation and minimal discrepancy in area measurements, thus underscoring its innovative and robust performance for semantic segmentation in lower extremity X-ray images. For comparison, U-Net achieved an IoU of 0.84 and a DC of 0.87, while V-Net showed an IoU of 0.87 and a DC of 0.91. Example images of the segmentation model output are displayed in the Appendix A.

### 3.3. Validation of AI-Based Segmentation Model for Muscle Mass Quantification

To validate the accuracy of the AI-based semantic segmentation model for muscle mass quantification, we compared the results from the segmentation model with those obtained from the Dual-Energy X-ray Absorptiometry (DEXA) technique. The segmentation model’s output was assessed against the DEXA skeletal muscle index, calculated as the appendicular lean mass divided by the square of the subject’s height.

As presented in Table 3, the Pearson correlation coefficient between the total muscle region estimated by the segmentation model and the DEXA-derived skeletal muscle index was 0.72 (*** *p* < 0.0001), suggesting a moderate to strong positive correlation. Additionally, the correlation between the total muscle region estimated by the model and DEXA-measured legs’ lean muscle mass was 0.66 (*** *p* < 0.0001). Individual muscle groups were further analyzed to evaluate the model’s performance across different regions. Table 3 details the Pearson correlation coefficients for the medial, lateral, gluteal, and calf muscle groups against both the skeletal muscle index and legs’ lean mass, indicating strong associations for most regions (e.g., medial 0.70 for skeletal muscle index and 0.57 for legs’ lean mass; lateral 0.66 for skeletal muscle index and 0.72 for legs’ lean mass; gluteal 0.65 for skeletal muscle index and 0.69 for legs’ lean mass, all with *** *p* < 0.0001). The calf muscle showed a weaker correlation (0.10 for skeletal muscle index, 0.14 for legs’ lean mass). Higher ‘r’ values indicate stronger associations between the X-ray-based segmentation results and DEXA-based indices.

As depicted in Figure 4A, the Bland–Altman analysis revealed a mean difference of -2653 g, with 95% limits of agreement ranging from −8158 g to 2851 g, between the leg lean mass measured by DEXA and the estimated muscle region mass derived from lower extremity X-rays. This plot visually illustrates the agreement, or lack thereof, between the two measurement methods. Furthermore, an ordinary least squares regression analysis between the estimated muscle region and legs’ lean mass produced a coefficient of 0.74 (t = 28.87, *p* < 0.0001), indicating a strong linear relationship between these two measurements, as shown in Figure 4B.

Finally, a paired t-test was conducted to assess the consistency between the muscle region estimated by the model and both the legs’ lean mass and skeletal muscle index. The results indicated a ** significant difference between the model’s estimated muscle region and legs’ lean mass (t-statistic = −6.34, *p* < 0.05), as well as between the estimated muscle region and skeletal muscle index ** (t-statistic = −23.77, *p* < 0.05).

## 4. Discussion

### 4.1. Study Overview

This study developed and validated an AI-based semantic segmentation model to quantify muscle mass from lower extremity X-ray images as a potential alternative to DEXA for sarcopenia assessment. We compared U-Net++, U-Net, and V-Net architectures and evaluated the correlation between X-ray-derived muscle measurements and DEXA-based indices in 66 patients. The model achieved strong performance metrics with an IoU of 0.93 ± 0.015 and Dice coefficient of 0.95 ± 0.008, demonstrating high accuracy in muscle boundary delineation. The correlation between X-ray-derived measurements and DEXA skeletal muscle index was r = 0.72 (*p* < 0.0001), indicating a statistically significant relationship between the two measurement methods.

### 4.2. Performance Evaluation and Clinical Significance

When compared to existing literature on muscle segmentation in lower extremity X-rays, our model’s performance is competitive yet distinct in its approach. A comparable study reported a numerically higher IoU value of 0.959 (95% CI 0.959–0.960) than our achieved IoU of 0.93 [25]. However, that model demonstrated reduced accuracy in critical anatomical regions, particularly the medial and gluteal fold areas where precise segmentation is challenging. In contrast, our model was specifically designed to analyze five distinct muscle regions including the gluteal region, achieving moderate to strong correlations with DEXA measurements for four of the five regions. This comprehensive regional analysis represents a significant advancement over previous approaches, as it provides detailed information about individual muscle groups rather than treating the lower extremity as a single unit. While the calf region showed weaker correlation in our analysis, the overall multi-regional approach offers superior clinical utility for targeted muscle assessment and sarcopenia evaluation compared to existing single-measurement methodologies.

The weaker correlation observed specifically in the calf muscle group (r = 0.10) is hypothesized to be attributable to the inherent limitations of 2D X-ray projection imaging. Anatomically, the major posterior calf muscles (such as the Soleus and Gastrocnemius) are significantly overlapped and obscured by the denser anterior bone structures, particularly the tibia, when viewed on a single projection. This substantial bone-muscle overlap complicates the precise delineation of muscle boundaries, potentially leading to misclassification of muscle tissue during both the ground truth labeling and the subsequent AI segmentation. Nonetheless, this localized technical constraint does not diminish the overall efficacy of the model, given the strong correlations achieved in the functionally critical muscle groups—the medial thigh (r = 0.70), lateral thigh (r = 0.66), and gluteal region (r = 0.65). The model’s robust performance in these three major regions underscores its potential as a reliable tool for sarcopenia assessment.

The correlation between X-ray-derived measurements and the DEXA skeletal muscle index was **statistically significant and moderate-to-strong (r = 0.72, * *p* < 0.0001). This relationship demonstrates that our AI-based method explains 52% of the variance (r^2^ = 0.52) in muscle mass, capturing a substantial, clinically meaningful portion of the variation detected by the gold standard technique. However, the remaining unexplained variance of 48% necessitates careful clinical interpretation, especially as it suggests additional factors may influence the relationship between the two measurements. This degree of correlation is particularly relevant for patients near sarcopenia diagnostic thresholds, where high measurement precision is critical for accurate clinical decision-making.

The Bland–Altman analysis provides essential insights into the clinical reliability and measurement agreement between our X-ray-based method and DEXA, particularly relevant for diagnostic applications. The systematic bias of −2653 g indicates that our model consistently underestimates muscle mass compared to DEXA measurements. While this systematic underestimation represents a predictable offset that could potentially be addressed through calibration protocols, it is an important consideration for diagnostic implementation. The 95% limits of agreement ranging from −8.158 kg to +2.851 kg represent the expected range of measurement differences between the two methods for individual patients.

For diagnostic applications, this measurement variability has important clinical implications, particularly for patients near sarcopenia diagnostic thresholds where differences within this range could influence classification decisions. However, the systematic nature of the bias suggests that with appropriate calibration and establishment of method-specific diagnostic cutoffs, our approach could serve as a reliable diagnostic tool rather than merely a screening instrument. The predictable offset indicates that the measurement differences are not random but follow a consistent pattern, which is advantageous for developing standardized diagnostic protocols. With further refinement and validation studies to establish optimal diagnostic thresholds specific to X-ray-derived measurements, this approach demonstrates significant potential for accurate sarcopenia diagnosis, particularly in clinical scenarios where DEXA limitations compromise diagnostic accuracy.

### 4.3. Advantages in Implant Patients

While DEXA is widely used for measuring bone mineral density and body composition, including muscle mass, the presence of implants can introduce significant inaccuracies. Metal implants, such as those used in joint replacements, can lead to overestimation of bone mineral content (BMC) and soft-tissue mass. Studies have shown that metal rods can inflate whole-body BMC measurements by 1.5–3% and significantly increase reported soft-tissue mass (*p* < 0.003) [26]. Similarly, breast implants have been found to increase whole-body BMC by 4.7% [27]. These inaccuracies extend to muscle mass measurements, with total knee replacements (TKR) causing overestimation of lean mass in affected limbs [28]. To address these issues, correction methods such as Automatic Metal Detection (AMD) processing and substitution protocols have been developed [29]. AMD processing has been shown to mitigate overestimation of muscle mass in patients with TKR, while substitution protocols involving the replacement of affected regions with measurements from unaffected contralateral sides have improved the accuracy of body composition assessments. These findings underscore the importance of accounting for implants when interpreting DEXA results, particularly in conditions like sarcopenia where precise muscle mass measurement is crucial. Clinicians must be aware of these potential inaccuracies to avoid misdiagnosis and ensure appropriate treatment decisions.

However, our AI-based muscle segmentation model using long extremity X-rays presents a promising solution to the limitations of DEXA in patients with implants. By directly analyzing image features to delineate muscle boundaries, rather than relying on X-ray attenuation, the AI model can potentially mitigate implant-related artifacts and provide more accurate muscle mass estimations. The model’s ability to adapt through learning, focus on specific muscle groups, and operate independently of density measurements offers advantages over DEXA in complex anatomical scenarios. This approach could lead to more consistent and reliable muscle mass quantification across diverse patient populations, including those with prosthetics or joint replacements. While further validation is necessary, particularly in patients with various implant types, our AI model shows potential for improving the accuracy of muscle mass assessment and longitudinal tracking in patients with implants, potentially enhancing the diagnosis and monitoring of conditions like sarcopenia.

One case in the cohort illustrates the limitations of using DEXA to diagnose sarcopenia in patients who have implants. The patient was diagnosed with an intertrochanteric fracture and underwent intramedullary surgery. DEXA scans were performed immediately after surgery and at a one-year follow-up. As Figure 5 shows, the one-year follow-up DEXA scan indicated a significant increase in the skeletal muscle index (from 5.15 to 6.31 kg/m^2^, 22.5% increased), incorrectly suggesting an improvement from severe sarcopenia to not severe sarcopenia. However, gait analysis results contradicted the DEXA findings and lower extremity X-rays revealed significant muscle area loss (from 77 to 71.5 cm^2^, 7.15% decreased) in one-year follow-up X-ray.

This case is particularly relevant because patients with the sarcopenia range are at higher risk for fractures and knee osteoarthritis, often requiring arthroplasty. These findings suggest that muscle estimation methods based on lower extremity X-ray segmentation method could be a viable alternative to DEXA for sarcopenia assessment in such cases. This approach may provide more accurate results, especially in patients who have undergone orthopedic surgeries that could potentially affect DEXA measurements.

### 4.4. Limitations and Future Directions

Our study has several limitations that provide clear directions for future research. First, there is significant selection bias due to the single-center design, as data was collected solely from a university medical center in South Korea with a cohort primarily composed of patients with hip fractures or hip-related orthopedic conditions. To address this limitation and enhance generalizability, multi-center studies incorporating diverse populations, age ranges, and various musculoskeletal conditions beyond hip fractures are essential for broader clinical applicability.

Second, while our model demonstrates statistical correlation with DEXA measurements (r = 0.72), the clinical significance and diagnostic implications require more robust evaluation, particularly given the wide limits of agreement (95% limits: −8.158kg to +2.851 kg) that may impact diagnostic decisions near sarcopenia thresholds. Future studies should conduct comprehensive diagnostic accuracy analyses, including sensitivity and specificity assessments for sarcopenia diagnosis, and investigate the impact on clinical decision-making and patient outcomes.

Third, our model focuses on muscle quantity rather than quality or function, which are crucial factors in comprehensive sarcopenia assessment. Integration of functional data such as gait analysis, grip strength, and Short Physical Performance Battery (SPPB) scores alongside muscle quantity measurements will provide more comprehensive musculoskeletal health evaluation.

Finally, our cross-sectional design and primary comparison with DEXA limit validation scope. Future research should include longitudinal studies to track muscle mass changes over time and comparisons with other imaging modalities like CT and MRI. These developments will enable broader clinical applications including early sarcopenia screening, treatment monitoring in rehabilitation programs, and personalized intervention strategies across diverse patient populations in musculoskeletal medicine.

## 5. Conclusions

Our AI-based semantic segmentation model achieved excellent performance (IoU = 0.93, Dice = 0.95) and strong correlation with DEXA (r = 0.72, *p* < 0.0001) in quantifying muscle mass from lower extremity X-rays. This approach offers significant advantages over DEXA, particularly in patients with artificial implants where conventional methods can lead to sarcopenia misdiagnosis. The model provides detailed muscle group information while mitigating implant-related measurement errors.

Despite limitations including single-center design, the need for clinical significance validation, and functional assessment integration, this study represents a significant advancement in muscle mass quantification. Future research should conduct multi-center validation studies, establish diagnostic accuracy metrics, and integrate functional assessments through longitudinal studies. This AI-driven approach has the potential to revolutionize sarcopenia diagnosis and monitoring, especially in complex patient populations where current methods fail.

## Figures and Tables

**Figure 1 healthcare-13-02488-f001:**
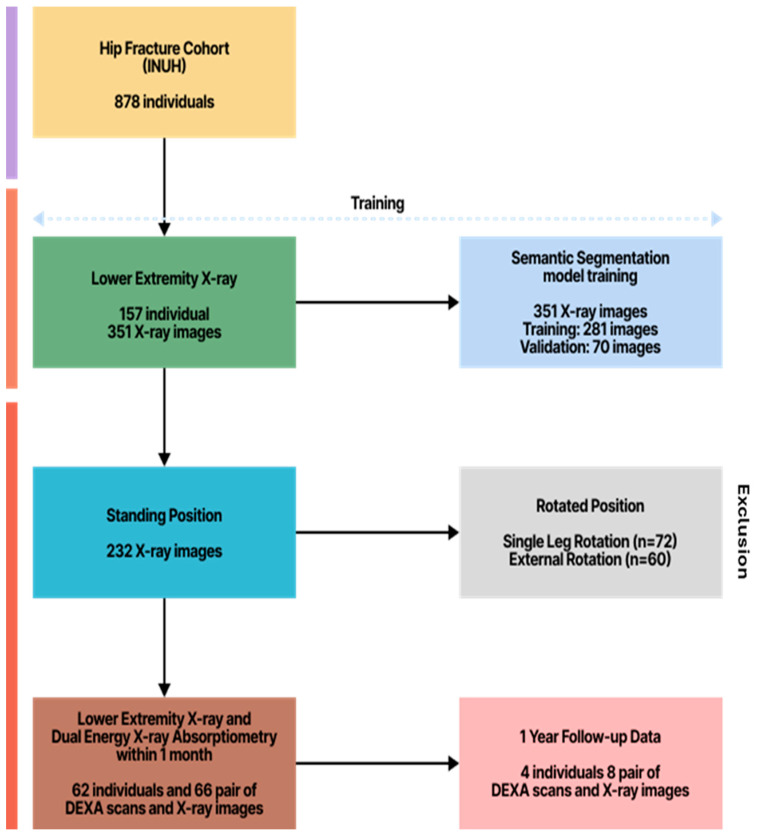
Patient selection flowchart. From 878 initial participants, 157 provided X-ray data (351 images). After excluding rotated images, 351 images were used for AI model development (281 training, 70 validation). Sixty-six paired DEXA-X-ray datasets from 62 participants were used for validation analysis.

**Figure 2 healthcare-13-02488-f002:**
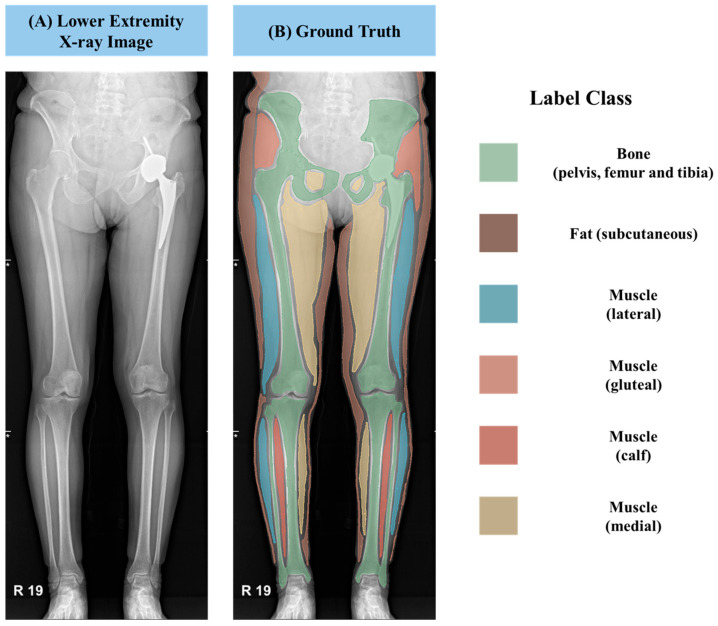
(**A**) Lower extremity X-ray image and (**B**) Corresponding ground-truth segmentation mask with color-coded labels for bone (pelvis, femur, and tibia), subcutaneous fat, and multiple muscle regions (lateral, posterior, gluteal, calf, medial). These labeled classes form the basis for subsequent segmentation and analysis.

**Figure 3 healthcare-13-02488-f003:**
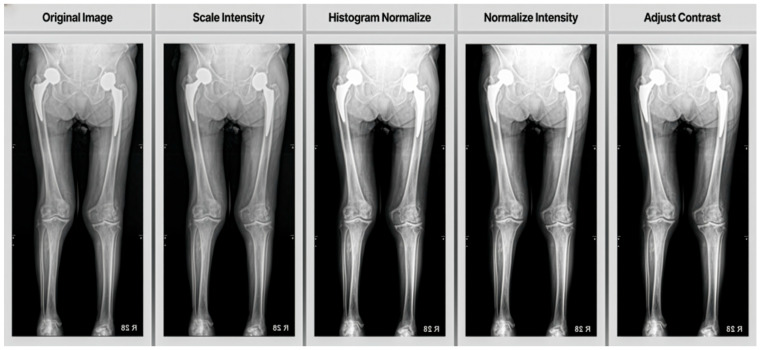
To prepare the X-ray images for segmentation, four key pre-processing steps were applied: intensity values were first scaled to a 0.0–1.0 range to ensure consistent brightness; histogram normalization redistributed pixel intensities for enhanced contrast; subsequent intensity normalization (subtracting the mean and dividing by the standard deviation) further standardized the data; and finally, gamma correction (γ = 1.6) adjusted contrast to emphasize important anatomical features.

**Figure 4 healthcare-13-02488-f004:**
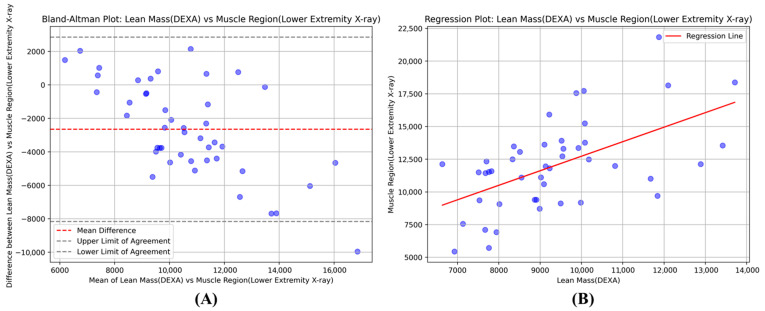
(**A**) The Bland–Altman plot illustrates the agreement between lean mass measured by DEXA and the muscle region identified on lower extremity X-rays, showing the mean difference (red dashed line) and upper/lower limits of agreement (black dashed lines). (**B**) The regression plot demonstrates the linear relationship between these two measurements, with the fitted regression line shown in red.

**Figure 5 healthcare-13-02488-f005:**
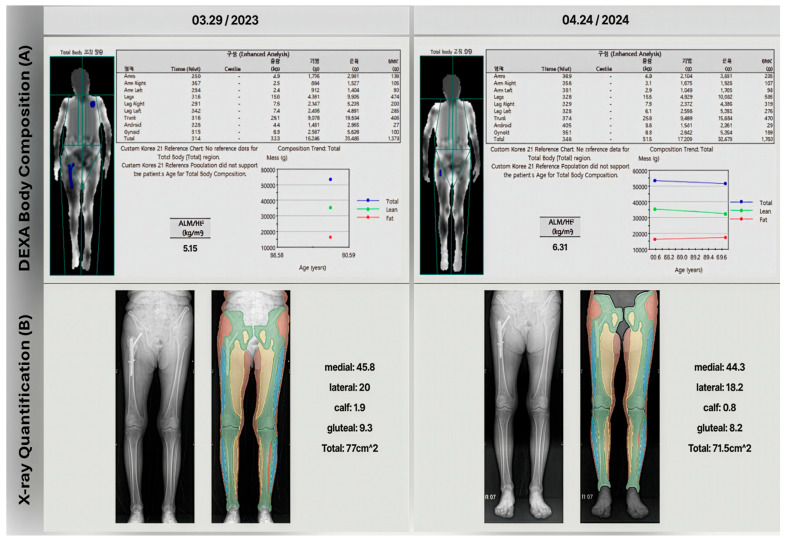
A single patient’s muscle measurements are shown at two time points (29 March 2023 and 24 April 2023). (**A**) displays DEXA body composition results with Korean column headers indicating “구성 (Composition)”, “영역 (Region)”, “총량 (Total mass)”, “지방 (Fat mass)”, and “근육 (Lean mass)”, while (**B**) presents corresponding X-ray-based muscle region quantifications. The color-coded segments illustrate changes in medial, lateral, and gluteal muscle areas over time.

**Table 1 healthcare-13-02488-t001:** Characteristics of the participants in DEXA measured groups.

		Female	Male	Total
Non-Sarcopenia	Number	19	-	19
	Age (SD)	79.9 (8.4)	-	79.9 (8.4)
	Height (SD)	156.0 (10.2)	-	156.0 (10.2)
	Weight (SD)	53.6 (9.6)	-	53.6 (9.6)
Sarcopenia	Number	7	1	8
	Age (SD)	79.2 (8.6)	83 (0)	79.6 (8.9)
	Height (SD)	153.3 (8.1)	169 (0)	156.3 (9.9)
	Weight (SD)	53.9 (7.7)	44 (0)	54.5 (9.4)
Severe Sarcopenia	Number	27	12	39
	Age (SD)	80.3 (8.3)	79.2 (7.7)	80.3 (8.3)
	Height (SD)	156.3 (10.1)	156.2 (10.5)	156.3 (10.0)
	Weight (SD)	54.6 (9.3)	54.7 (9.5)	53.6 (9.4)

Demographic data (number of participants, age, height, and weight [mean ± SD]) for female, male, and total participants are presented across non-sarcopenia, sarcopenia, and severe sarcopenia groups, as determined by DEXA measurements.

**Table 2 healthcare-13-02488-t002:** Performance of Semantic Segmentation Models in Validation Datasets.

	IoU	DC	AD	HD	RAAD (%)
U-Net	0.84	0.87	3.14	7.21	11.2
V-Net	0.87	0.91	2.87	6.50	7.13
U-Net++	0.93	0.95	0.89	1.92	1.80

U-Net, V-Net, and U-Net++ were evaluated using IoU, Dice Coefficient, Average Distance, Hausdorff Distance, and RAAD. U-Net++ achieved the highest IoU (0.93) and Dice (0.95) alongside the lowest Average Distance (0.89), Hausdorff Distance (1.92), and RAAD (1.80%).

**Table 3 healthcare-13-02488-t003:** Pearson’s correlation analysis between legs lean mass and skeletal muscle index from DEXA and muscle region from lower extremity X-ray.

	Skeletal Muscle Index (kg/m^2^)		Legs Lean Mass (g)	
	Correlation Coefficient	*p*-Value	Correlation Coefficient	*p*-Value
Medial	0.70	*** <0.0001	0.57	*** <0.0001
Lateral	0.66	*** <0.0001	0.72	*** <0.0001
Gluteal	0.65	*** <0.0001	0.69	*** <0.0001
Calf	0.10	0.52	0.14	0.36
Total	0.72	*** <0.0001	0.66	*** <0.0001

***: *p* < 0.001. Pearson’s correlation coefficients (r) and *p*-values comparing each muscle region identified by X-ray segmentation (medial, lateral, gluteal, calf, and total) with skeletal muscle index and legs lean mass derived from DEXA. Higher r values indicate stronger associations between the X-ray-based segmentation results and DEXA-based indices.

## Data Availability

The original contributions presented in this study are included in the article. Further inquiries can be directed to the corresponding author.

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
