# Peer review of "AI-Enhanced Lower Extremity X-Ray Segmentation: A Promising Tool for Sarcopenia Diagnosis"

_healthcare, 2025, doi:10.3390/healthcare13192488_

Round 1
Reviewer 1 Report
Comments and Suggestions for Authors
Sarcopenia, characterized by progressive loss of skeletal muscle mass and strength, is a significant chronic condition in older adults, reducing physical function, quality of life, and life expectancy. Although recent guidelines, such as those from the Global Leadership Initiative on Sarcopenia (GLIS), recommend precise assessments, traditional measurement methods like Dual-energy X-ray absorptiometry (DEXA) are expensive and often inaccessible in primary care.
AUTHORS aimed to develop and validate an AI-driven auto-segmentation model for muscle mass assessment using long X-rays as a cost-effective alternative to DEXA.
A total of 351 lower extremity X-ray images from the Real Hip Cohort in South Korea were collected. AI-based semantic segmentation models, including U-Net, V-Net, and U-Net++, were trained and validated on this dataset.
Model performance was assessed using Intersection over Union (IoU) and Dice Similarity Coefficient (DC). Correlation with DEXA-derived skeletal muscle index was evaluated using Pearson correlation and Bland-Altman analysis.
In sum AUTHORS analyzed data from 157 patients (117 females, 40 males; mean age 77.1 years). The U-Net++ architecture achieved the best segmentation performance with an IoU of 0.93 and DC of 0.95. Pearson correlation demonstrated a moderate to strong positive correlation between the AI model’s muscle estimates and DEXA results (r = 0.72, ***p < 0.0001). Regression analysis showed a coefficient of 0.74 (t = 28.87, ***p < 0.0001), indicating good agreement with reference measurements.
THEY concluded that:- THEIR study successfully developed and validated an AI-driven auto-segmentation model for estimating muscle mass from long X-rays. -The model provides a cost-effective, accessible alternative to DEXA, with potential to improve sarcopenia diagnosis and management in community and primary care settings.- Future work will refine the model and explore its application to additional muscle groups and related conditions.
The study is interesting and has merit.
I have the following comments for the authors:
1) The abstract is well written; perhaps fewer acronyms and less data would make it easier to read.
2) The citation on line 48 is non-standard; in some cases, the citations are grouped as in line 53.
3) The introduction is very brief with few references. It is suggested that both the content and the references be expanded, taking care to first illustrate the medical problem and then the approach to the literature.
4) The study design reported in the methods has as its strengths the enthusiasm and broad scope of the study, which also includes several models with interesting theoretical approaches. Its only weakness is the lack of a brief introductory summary, which risks the reader getting lost.
5) The results are interesting. The positive aspect is the innovation and richness of the analysis. In my opinion, there are two weaknesses. The first is that they could be organized by themes, in line with the study design, which includes several models. The second is that the medical application should also be highlighted: I wonder if it wouldn't be appropriate to move Figure 5 and include some application images to enhance the value. Keep in mind that the journal is healthcare, not computer science.Figure 1 (this applies to all figures) should be fully described in the body of the text.
6) Data is usually not included in the conclusions.
7) It is suggested that a paragraph with suggestions for future developments be included.
In general, give adequate space to the medical application both in the introduction and in the other parts
Reviewer 2 Report
Comments and Suggestions for Authors
Dear editors,
Thanks for the opportunity to review the manuscript «AI-Enhanced Lower Extremity X-ray Segmentation: A Promising Tool for Sarcopenia Diagnosis».
This manuscript presents a contribution to the field of sarcopenia diagnosis by introducing a novel, AI-driven auto-segmentation model for muscle mass estimation from lower extremity X-rays. The authors' stated objective is to provide a more accessible and cost-effective alternative to the current gold standard, Dual-energy X-ray absorptiometry (DE bXA). The study outlines a complex methodology, including the use of a comprehensive dataset from a specific cohort, the application of deep learning architectures like U-Net++, and a validation process against DEXA measurements. The authors conclude that this tool holds significant promise for enhancing the detection and management of sarcopenia.
I enjoyed reading the article, and I believe it can be of interest. However, there are several aspects that I would recommend revising. These are the following:
The introduction of the abstract is too lengthy. The methods section, however, is missing key details. For example, it should state the type of study you conducted before you mention how many images were taken.
While the authors justify their work by stating the need for more cost-effective diagnostic methods, the introduction lacks a more detailed explanation of this need. You need to provide a more compelling justification for why using AI is specifically beneficial in this particular case. The authors should elaborate on the unique advantages of AI in this context.
The methods section is disorganized.
In the manuscript's methods section, the authors get straight to the training process in the first sentence. They likely did this to keep the text short.
I recommend starting with a broader overview and then getting more specific. First, describe the type of study you conducted. Then, you can talk about the setting and the eligible population.
The second paragraph abruptly introduces ethical considerations before returning to the methodology.
The authors also discuss the manual segmentation of images, but provide almost no details on how it was performed. This lack of detail makes the work difficult to replicate and verify. They also don't specify who they consider to be an "expert."
A similar issue occurs when they discuss image pre-processing. They mention the aspects they handled, but don't explain how or provide a protocol. This could mean the process was not standardized, depended on a particular expert, or was subjective.
The authors need to provide a clearer explanation of these aspects to reduce subjectivity and enhance the study's reproducibility. Following a systematic and reproducible methodology is crucial, especially in this type of research, as factors such as AI training can be difficult to control and may vary across different studies. Therefore, it's essential that you are very systematic when you describe your methods. If you aren't, other researchers won't be able to replicate your work.
The authors are also missing a more detailed explanation of the eligible population and the inclusion and exclusion criteria. These are essential for a better understanding of the training process and for exact replication.
In the methods section, subsection 2.4, the authors mention the architectures they used. While this information is helpful for context, it hinders the readability and comprehension of the methodology.
I think these structures should be summarized in the introduction to provide context, allowing the methods section to focus solely on the specific actions taken. Alternatively, if they remain in the methods section, they should be condensed as much as possible to avoid disrupting the flow of the text.
In the results section, which only includes data from images, a table with basic patient demographics should also be included. This information is always helpful because demographic factors can sometimes bias the results. This table would help readers analyze the external validity of your findings.
The discussion starts by jumping right into a conclusion. It would be much more organized if the authors began with a one- or two-line summary of their work. Then, they should compare their results to existing literature and discuss the study's limitations. This structure would allow them to present their conclusions at the end, based on all the evidence. Starting with the conclusion gives the impression that it was preconceived before the study even began.
The clinical significance of the model's performance needs to be addressed more explicitly and robustly. Although a Pearson correlation coefficient of 0.72 is statistically significant, the clinical implications of this level of agreement have not been fully explored.
The Bland-Altman analysis, although mentioned, is not thoroughly presented or discussed. A discussion of the bias and limits of agreement between the two methods is essential for clinicians to understand the model's reliability in a real-world setting.
In the limitations section, the authors could provide a clearer explanation of the selection bias: the study relies on a cohort of patients from a single medical center in South Korea, primarily composed of individuals with hip fractures.
While the topic is complex and innovative, the authors use only 15 references. I believe that a more comprehensive literature search could likely enrich the introduction and discussion sections by including relevant, albeit not identical, work. Although the specific focus of this paper is narrow, the use of trained AI for image diagnosis and interpretation is a well-established and expansive field of study. Therefore, a wealth of literature is available on the more general aspects of this work that could be drawn upon.
Reviewer 3 Report
Comments and Suggestions for Authors
Thank you very much for the invitation to review this work. The manuscript entitled “AI-Enhanced Lower Extremity X-ray Segmentation: A Promising Tool for Sarcopenia Diagnosis” addresses an important and timely topic: the development of AI-based muscle segmentation from long X-rays as an alternative to DEXA for sarcopenia assessment. The study demonstrates promising accuracy, especially with the U-Net++ architecture, and highlights clinical utility in settings where DEXA is not accessible. However, several methodological, clinical, and reporting issues must be addressed before the manuscript can be considered for publication.
Major Comments
- Study Design and Cohort Limitations
- The dataset is derived from a single-center cohort in South Korea, with a relatively small number of DEXA comparisons (66 scans). This limits external validity and generalizability. The authors should more explicitly discuss how this affects model robustness across different ethnicities, body types, and imaging systems.
- Please clarify whether the dataset includes only elderly hip fracture patients, or if there was broader inclusion. This strongly influences representativeness for sarcopenia screening in general populations.
- Methodological Transparency
- Details of patient selection and exclusion criteria are insufficient. Were patients with metal implants included in the training set? Were poor-quality images excluded?
- The study mentions various preprocessing techniques (intensity scaling, histogram normalization, gamma correction). Authors should provide a rationale for why these specific methods were chosen and whether they could introduce bias.
- Please justify the choice of 2D models only (U-Net, V-Net, U-Net++). Given the volumetric nature of CT/MRI and the potential of pseudo-3D X-ray reconstructions, was 3D segmentation considered?
- Clinical Relevance and Interpretation
- While statistical correlations with DEXA are reported, the clinical thresholds for sarcopenia diagnosis are not well addressed. How would the AI-derived measures be integrated into current GLIS, AWGS, or EWGSOP2 diagnostic workflows?
- The calf region results were poor (r = 0.10). This is clinically relevant because calf circumference is often used as a surrogate for muscle mass. The authors should discuss why the model fails in this region and whether anatomical factors or data limitations contributed.
- A case example with implants is included, but more structured evidence is needed. Can the authors provide subgroup analyses for patients with and without implants?
- Statistical Analysis
- Bland–Altman plots show wide limits of agreement (–8,158 g to +2,851 g), which raise concerns about clinical applicability. The discussion should explicitly state whether such variance is acceptable for diagnostic use.
- The paired t-test results indicate significant differences between AI-estimated and DEXA-derived values. This undermines the claim of interchangeability. The authors should clarify whether the model is intended as a screening tool rather than a replacement for DEXA.
- Presentation and Reporting
- Figures and tables need improvement for clarity. For example, Figure 4 (Bland–Altman) should include percentage error limits and clinical acceptability ranges.
- Supplementary images showing successful and failed segmentations across different muscle groups would strengthen the paper.
- The manuscript contains minor inconsistencies in reporting funding and author contributions (e.g., “Funding: This research received no external funding” contradicts listed grant numbers). These must be corrected.
- Future Directions
- The authors mention future work but do not propose clear strategies. Please outline specific next steps: multicenter validation, integration with functional outcomes (grip strength, gait speed), or hybrid imaging with ultrasound.
Comments on the Quality of English Language
Need a native speaker to proof it before publication.
Reviewer 4 Report
Comments and Suggestions for Authors
Dear Editor and Authors,
The authors develop a semantic-segmentation pipeline on full-length lower-extremity radiographs to estimate lower-limb muscle and compare these estimates against DEXA-derived skeletal muscle index (SMI). A single-centre dataset from a “Real Hip Cohort” in South Korea was assembled (157 patients; 351 radiographs; 66 paired DEXA scans from 62 patients). U-Net, V-Net, and U-Net++ were trained (281 images for training; 70 for validation); U-Net++ performed best (reported IoU 0.93, Dice 0.95 on validation). Correlation with DEXA was “moderate-to-strong” overall (e.g., r≈0.72 for total muscle region; region-wise correlations 0.65–0.70 for thigh compartments but poor for calf, r≈0.10, n.s.). Bland–Altman analysis showed wide limits of agreement (≈ −8.16 to +2.85 kg vs DEXA leg lean mass), and paired tests indicated significant mean differences between modalities. The paper argues that AI segmentation from radiographs may offer a cost-effective, implant-artifact-resistant alternative to DEXA in primary and community care.
-Methods cite Spearman (and later Pearson) correlations; results report Pearson. Segmentation performance is reported as IoU 0.93/Dice 0.95 in the results table, but the discussion mentions IoU 0.91 ± 0.015—these figures conflict. Units and terminology (“volume” derived from 2D pixel area) are inconsistent; conversion to “physical volume” from single-view X-rays lacks a thickness model, so the quantity is effectively area, not volume.
-Two experts provided annotations, but inter-rater agreement and adjudication procedures are not reported.
*****This is a highly technical article and requires a multidisciplinary approach for its evaluation. Therefore, I believe it would be beneficial for the article to be reviewed by a software engineer with a core focus and interest in AI, as well as a statistician, in addition to physicians. Therefore, I request that the editor assign these reviewers from relevant disciplines to evaluate.*****
In short, as an orthopedic surgeon, I thoroughly enjoyed the article. I believe it will contribute significantly to the literature.
Reviewer 5 Report
Comments and Suggestions for Authors
The publication presents a study aiming to develop and validate an AI-driven auto-segmentation model for muscle mass assessment using long X-rays. The goal is to establish a standardized protocol for image acquisition and to compare the model’s accuracy and reliability with the current gold standard, whole-body Dual-Energy X-ray Absorptiometry (DEXA) measurements. The methodology and statistical analysis are presented in detail, and the authors provide some comparisons with previously published data.
- However, the study sample is unevenly distributed, with 351 lower extremity X-ray images versus 66 DEXA scans, which may affect the results.
- The authors claim that their approach is cost-effective, but no supporting evidence or calculations are provided.
- In addition, important aspects are not addressed, such as time requirements for performing and evaluating the analyses, the complexity of interpretation, the need for specialized training, and the radiation exposure involved comparing both methods.
- Regarding presentation, the Figure 2 is not very clear and should be improved for better readability.
Overall, the manuscript provides a promising approach and could make a meaningful contribution to the field. After clarifying the above-mentioned issues, I recommend the article for publication.
Round 2
Reviewer 3 Report
Comments and Suggestions for Authors
Thank you very much for responding and revising the manuscript as the reviewers commented. I accept the present form of publication.